

# Quantifying the contribution of forcing and three prominent modes of variability on historical climate

Andrew P. Schurer[1]\*, Gabriele C. Hegerl[1], Hugues Goosse[2], Massimo A. Bollasina[1], Matthew H. England[3], Michael J. Mineter[1], Doug M. Smith[4] and Simon F. B. Tett[1]

[1]School of GeoSciences, University of Edinburgh, Edinburgh, EH9 3JW, United Kingdom

[2]Université Catholique de Louvain, Georges Lemaître Centre for Earth and Climate Research, Earth and Life Institute, Louvain-La-Neuve, B-1348, Belgium

[3]Climate Change Research Centre and ARC Centre for Excellence in Antarctic Science, University of New South Wales, New South Wales 2052, Australia

[4]Met Office Hadley Centre, Exeter, EX1 3PB, United Kingdom

*Correspondence to*: Andrew Schurer (a.schurer@ed.ac.uk)

**Abstract.** Climate models can produce accurate representations of the most important modes of climate variability, but they cannot be expected to follow their observed time-evolution. This makes direct comparison of simulated and observed variability difficult, and creates uncertainty in estimates of forced change. We investigate the role of three modes of climate variability, the North Atlantic Oscillation, El-Niño Southern Oscillation and the Southern Annular Mode, as pacemakers of climate variability since 1781, evaluating where their evolution masks or enhances forced climate trends. We use particle filter data assimilation to constrain the observed variability in a global climate model without nudging, producing a near free running model simulation with the time-evolution of these modes similar to those observed. Since the climate model also contains external forcings, these simulations, in combination with model experiments with identical forcing but no assimilation, can be used to compare the forced response to the effect of the three modes assimilated, and evaluate to what extent these are confounded with the forced response. The assimilated model is significantly closer than the "forcing only" simulations to annual temperature and precipitation observations over many regions, in particular the tropics, the north Atlantic and Europe. The results indicate where initialized simulations that track these modes could be expected to show additional skill. Assimilating the three modes cannot explain the large discrepancy previously found between observed and modelled variability in the southern extra-tropics but constraining the El-Niño Southern Oscillation reconciles simulated global cooling with that observed after volcanic eruptions.

## 1 Introduction

Understanding the causes of observed climate change is crucial not only to gain knowledge on the past but also to improve projections of future change. It is common to split the drivers of climate change into two broad categories, external forcings, which could have natural or anthropogenic origins, and internal variability (see e.g. Hegerl and Zwiers 2011, Eyring et al 2021). Examples of external forcings include changes in greenhouse gases and anthropogenic aerosols as well as volcanic eruptions and changes in solar radiation. Internal variability is caused by chaotic fluctuations generated internally by the



climate system. Separating the external forcing in observed climate change from the internal variability background, is often referred to as detection and attribution (Hegerl and Zwiers 2011). Uncertainty in the model response to external forcing and
internal variability are responsible for much of the uncertainty in future projections (see Lehner et al., 2020).

The combined effect of all external forcings or combinations of different forcings can potentially be determined by simulating them in climate models (e.g. Eyring et al. 2016) but these simulations will also include internal variability. For detection and attribution studies, which are primarily focused on the external forced component, the effect of the internal variability on the model signal (often called the fingerprint of change) is reduced by averaging over large ensembles (Gillett et al. 2021). These
studies typically treat internal variability as a statistical construct with properties calculated from pre-industrial simulations (piControl - model simulations which do not include any forcings) (see e.g. Allen & Stott, 2003, Ribes et al. 2017). Results can not only be used to understand the past but can also be used to constrain future projections (Kettleborough et al. 2007, Brunner et al. 2020). Other studies use large ensembles to study the combined role of internal variability and forcing (Deser et al. 2020) and changes in climate variability (Olonscheck et al. 2021). Since each simulation contains one realisation of internal
variability the ensemble spread can be used to estimate the possible range of future climates. While large ensembles can be used to estimate a range of plausible pasts, there is zero probability that any one model simulation will have same detailed evolution as observed, due to chaotic behaviour in the climate system. Therefore determining internal variability and its contribution to past change is challenging and is typically done by removing the forced component from observations either using model results (Hegerl et al. 2018, Friedman et al. 2020) or by de-trending (Knight 2005).

Since climate variability is very complex, acting across multiple time and space scales it is common to only study different aspects of it by isolating distinct modes of variability such as El-Niño Southern Oscillation (ENSO) (Timmermann et al. 2018) or the North Atlantic Oscillation (NAO) (Hurrell et al. 2003). These are typically assumed to be manifestations of internal variability, and are found in piControl simulations, however they can also be influenced by external forcings (e.g Smith et al. 2020; Khodri et al. 2017). By focussing on a particular expression of variability in this way it is possible to study various
aspects of its effect on climate. This includes: estimating the past effect of a particular modes of variability (Iles & Hegerl 2017, Hartmann et al 2013); determining whether it's possible to predict how the mode will evolve over the next couple of years (Chen and Cane 2008, Smith et al. 2020); and predicting if this pattern or amplitude is likely to change into the future (Collins et al 2013, Cai et al. 2015). Some studies have attempted to simulate past changes by perturbing climate simulations to mimic the observed evolution of different climate modes. Kosaka and Xie (2013) prescribed observed SSTs over the central
Pacific to force a simulation to follow the observed ENSO, while Delworth and Zeng (2016) and Delworth et al. (2016) prescribed heat flux anomalies to mimic the effect of the NAO on the Atlantic Ocean.

In this study, we adopt an existing data assimilation technique (the particle filter, van Leeuwen 2009) that has already been used in a number of studies (for example Goosse et al. 2012) to assimilate three major modes of internal variability in a historical model simulation (starting in 1781) which also accounts for all the most important external forcings. In this way the
simulations will have both the external forced component of past change, in addition to the correct evolution of three of the most important modes of internal variability, without the need to impose any additional fluxes or SSTs. In combination with



simulations without any external forcings (piControl simulations) and simulations with realistic external forcings but non-assimilated internal variability, the roles of the different drivers of past change can be evaluated. In section 2 we will introduce the experimental set-up, while section 3 analyses how well it performs compared to observed climate. The article will then finish with a brief discussion and conclusions section.

## 2 Experiment Setup

We use a set of simulations with the coupled atmosphere–ocean model HadCM3 (Pope et al. 2000, Gordon et al. 2000). The atmosphere has a horizontal resolution of 3.75° × 2.5° in longitude and latitude with 19 vertical levels. The ocean model has a resolution of 1.25° × 1.25° with 20 levels. Despite the relatively low resolution and the no longer state-of-the-art physics modules, we expect the model results to be meaningful, as the model has shown a high level of skill, putting it consistently among the top half of CMIP5 models (see e.g. Flato et al. 2013, Sanderson et al. 2015, Knutti et al. 2013) and performs well compared to CMIP6 models (Tett et al. 2022). The model is sufficiently fast and efficient to run the large quantity of model years required for this study. The forcings used are those described in Schurer et al. (2014), with the exception of the anthropogenic aerosol and ozone emissions which are updated to the CMIP5 protocol. Two sets of experiments have been run covering the period from 1781-2008.

The first set is a 10 member all forcing ensemble from 1781 to 2008 with time varying greenhouse gas and aerosol emissions, land cover changes as well as natural forcings (volcanic, solar, orbital), initialised from existing simulations (the four ALL forced simulations and 4 NoAER simulations presented in Schurer et al., 2014). These simulations are those described and analysed in Brönnimann et al. (2019) and will be referred to as "forcing-only" in the rest of the article.

For the second set of experiments we use the same model set-up with the same forcings as described above, initialised from the same initial conditions, but using a particle filtering method without nudging (van Leeuwen 2009). This technique has already been successfully used in several different studies (e.g. Goosse et al. 2012). It results in a physically consistent near-continuous simulation which tracks a set of chosen indices, which due to the nature of the filter need to have a low dimensionality (usually this would mean only using a small number of indices or strongly filtered spatial fields). In this case the North Atlantic Oscillation (NAO), the El Niño–Southern Oscillation (ENSO), and the Southern Annular Mode (SAM) were chosen as they have been found by numerous studies to have a dominant role on large-scale atmospheric variability (see e.g. Eyring et al 2021). Our simulations cannot be expected to be as close to the "truth" as the Kalman filter or 4D-var approaches commonly employed by reanalyses for example the 20[th] century reanalysis (Compo et al. 2011), the last millennium reanalysis (Hakim et al., 2016) and the EFK paleo-reanalysis (Franke et al. 2017) which are assimilating much more information such as surface temperature and pressure observations. Instead this set-up will produce an analysis which will behave like a free running model that happens to share relatively closely (but not perfectly) these three modes of variability with the observations. We can use this technique to see what effect simulating a realistic variability (on a seasonal scale) has on annual and decadal variability, and what aspects of decadal climate evolution are improved compared to fully free running



model simulations with the same external forcing (but without the data-assimilation). This analysis will also identify regions

where despite the assimilated variability observed changes remain inconsistent with the model simulation which could be

interpreted as a possible indicator of model error or data uncertainty.

The particle-filter set-up is based on that described in Dubinkina et al. (2011). In our analysis, 50 model simulations are started

in 1781 from initial conditions taken from the same simulations as the ensemble of 10 transient simulations. Every year, the

simulations are stopped on the 1st April and the likelihood of each of the 50 simulations (often called "particles") is calculated

based on the observations of the chosen three indices over the preceding 12 months. Another set of 50 simulations is then

generated with initial conditions taken from the end of these existing 50 simulations, sampled according to their likelihood,

with the lowest likelihood particles stopped and multiple new simulations initialised from the higher likelihood particles. A

tiny perturbation is made to the atmosphere of each of the initial conditions and 50 new simulations are run for the next year.

The likelihood, $p$, of a particle, $\psi$, given the observations, $d$, is based on a Gaussian density,

$$p(d|\psi) = K^{-1} exp\left[-\frac{1}{2}(d - H(\psi))^T C^{-1}(d - H(\psi))\right] \quad (1)$$

where $K$ is a normalization constant. $H$ is an observational operator which maps the model output onto the observational phase

space and $C$ is the error covariance matrix which here takes into account the observational uncertainty (see Dubinkina et al.

2011 for more details). The likelihood is thus dependent both on the closeness of each particle to the observed index but also

on the relative uncertainty of each index. The number of particles to spawn from each individual simulations is proportional

to its likelihood following an iterative process described in Dubinkina et al. (2011) with the only modification that here we

only allow each particle to get re-spawned a maximum of 20 times, so maintaining a spread of initial conditions for each new

iteration.

A schematic, shown in Fig. 1, illustrates schematically the particle filter technique used here, with just 10 particles and one

assimilated index. This toy-example shows the performance for 10 assimilation steps where at each step the initial conditions

for the next step are chosen from the previous particles based on the likelihood calculated with equation 1. The assimilated

product can be derived at each step for any variable by calculating a likelihood-weighted mean of all simulations using the

likelihood in Eq.1. We term this the weighted mean. A sequence of continuous particles can be found by time-reversing the

trajectory by linking each particle to its parent (Fig. 1). After only a few assimilation steps back it is inevitable that they will

all converge onto one particle and will follow the same sequence thereafter. We term this the continuous particle and for

simplicity it will be derived by only following the most likely particle at the final assimilation step backwards. In the particle

filter experiment this pseudo-continuous simulation will also exist (with only tiny a perturbation at machine precision being

applied to the atmosphere at each restart step). In theory, such a simulation could also exist without any assimilation step at

all, but this would require a prohibitively large number of simulations. Unless stated differently results will be shown for the

weighted mean and will be referred to as the "DA" (data assimilation) model simulation.





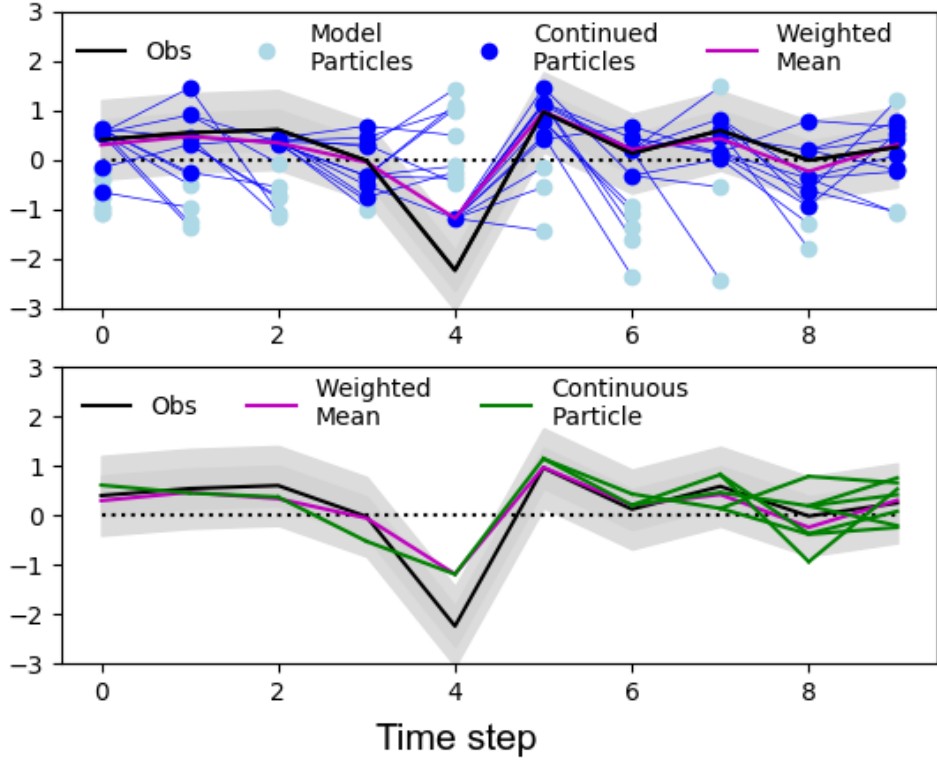

**Figure 1 – Schematic showing the performance of the particle filter.** *Values are only illustrative and do not represent any particular climate variable. Observations – black with grey uncertainties. Model simulations blue, simulations that are continued use a darker blue. Weighted mean purple, continuous simulations green (note that this quickly converges onto a single simulation as you follow the simulations back in time from the end of the experiment).*

The indices chosen are shown in Fig. 2. The data used to constrain the data assimilation simulation is intended to be the best available for that particular index. Hence, instrumentally observed indices are used when available and proxy reconstructions when they are not. The consequence of this is that the uncertainty of the target index changes through time, as does the relative uncertainty between the indicies and thus the constraint applied to the simulations.

For the NAO index the metric used is the mean of the NAO index over the period December to March and is calculated as the standardised difference between the standardised pressure of the Icelandic low and the Azores high, as in Luterbacher et al. (2001). For ENSO the metric used is temperature anomaly over the Nino3.4 region (5N-5S, 170W-120W). For 1781-1881 this is the mean over the months November to February and is obtained by the mean of the proxy reconstructions of Li et al. (2013) and Emile-Geay et al. (2013). In 1882 the metric changes to take advantage of the more reliable instrumental observations (HadSST2, Kennedy et al. 2011a; 2011b), with two periods used, the mean of the Nino3.4 index from April to September and the mean of the index from October to March, to find particles that follow the ENSO evolution during the whole year. The SAM index is initially the annual reconstruction of Abram et al. (2014). Given that the assimilation occurs on the 1st April



every year, it means that the first part of the year used to calculate the SAM index is from the previous assimilation step. In 1957 the filter switches to use the instrumental Marshall index (Marshall 2003). Given that monthly data exists the mean of the period April through to March is used to coincide with the assimilation time-step.

Given the finite number of particles (50), none of the individual simulations is capable of perfectly matching all three indices simultaneously. Which index will be given the most weight will be determined by the uncertainty of that particular index (see eq. 1). The uncertainty associated with each of the indices is shown in the supplement (Fig. S1). The uncertainties in the proxy reconstructions for the ENSO and particularly the SAM are much larger than the NAO instrumental uncertainty. Thus, the analysis initially tracks the NAO better as the filter preferentially tries to fit this index, as shown by correlations between the

assimilated and observed index in Fig. 3 (correlations for the "continuous particle" shown in the supplement Fig. S2). Uncertainties in ENSO and then the SAM decrease during the instrumental period (substantially in the case of the SAM) thus the correlation to these indices improves (with the match to the NAO noticeably degrading after 1950). The performance of the filter is also reflected in the number of particles that are re-spawned at each iteration (Fig. S3). This is about 17 till 1881, dropping to 10 when the ENSO instrumental data is used, and below 5 when the SAM instrumental data starts in 1957. As the

number of particles which match the observed indices, within their associated uncertainty decreases, so more particles are discontinued and more spawned from the remaining particles. Towards the end of the simulation the filter often only re-spawns from 3 particles, the minimum allowed. It is important to note that improvement in the agreement on an annual scale is not necessarily related to agreement on a decadal scale as modelled values consistently higher or lower than those observed can lead to large decadal disagreement, for example in the decadally-smoothed ENSO index in the mid-20th-century (Fig. 3).

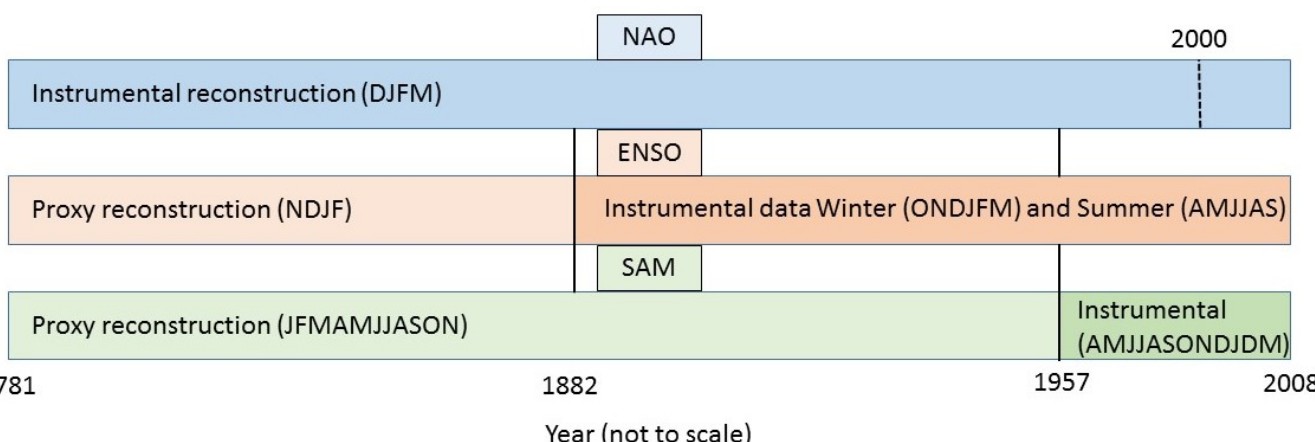


**Figure 2 – Schematic diagram of target indices –** *NAO instrumental reconstruction uses* (Luterbacher et al. 2001) *up until the year 2000 and then the 20th Century reanalysis* (Compo et al. 2011) *thereafter. ENSO proxy reconstruction uses a combination of* Emile-Geay et al. (2013) *and* Li et al. (2013) *until 1882 and then HadSST2* (Kennedy et al. 2011a, Kennedy et al. 2011b) *thereafter. SAM proxy reconstruction used is* Abram et al. (2014) *until 1957 and then the Marshall Index* (Marshall

2003) *thereafter.*





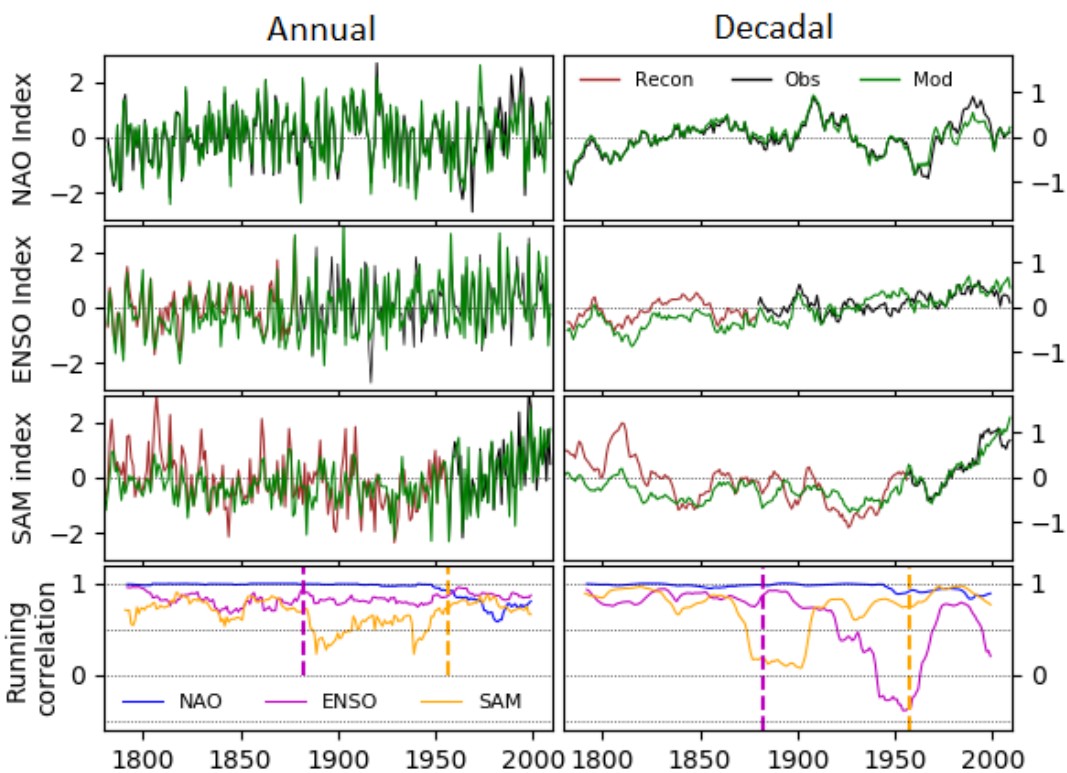

**Figure 3 – Performance of filter –** Top three rows – time-series of target indices, *NAO (DJFM, climatology period 1782-2008), ENSO (NDJF, climatology period 1882-1992), SAM (Annual, climatology period 1957-1995) for the weighted mean analysis (green), Reconstruction (orange), Instrumental Observations (black). Left column - annual values, right column – annual values smoothed by an 11 year running mean. Bottom panel shows, left – running 20- year correlation for the annual value, right – 40-year correlations for the decadal values for the NAO (blue), ENSO (purple), SAM (orange), vertical dashed line show the switch between proxy and instrumental values for ENSO and SAM.*

## 3 Results

### 3.1 Performance Validation

The performance of the particle filter experiment is assessed by analysing how well it reproduces the spatial patterns associated with the assimilated indices. The NAO, ENSO and SAM indicies were calculated in the observational record and in the particle filter simulation and in Fig. 4 the sea level pressure fields are regressed on these indicies. A comparison of the model pattern (right panels) to the observational counterparts (left panels) can be used to assess whether the HadCM3 climate model is capable of accurately simulating the spatial patterns of these modes of variability. In general the model performs well, however

there are some noticeable biases in particular in the temperature response to ENSO, which is known to extend too far west in



HadCM3 (see also Collins, Tett, and Cooper 2001), and in the temperature response to the SAM (although data availability prevents a comparison over the Antarctic continent). The central panels show the DA simulation fields regressed on the observed index. This represents a test of the assimilation procedure. If the particle filter was performing perfectly the modelled indices would be identical to the observed indices and consequently the central and right panels would be identical. This is

almost the case, (although the patterns using the observed index are slightly weaker), which is reflection of the fact that the modelled indices follow the instrumental observations reasonably well although not perfectly (Fig. 3). Regressing the DA simulation fields on the proxy SAM reconstruction results in a much weaker pattern (not shown) reflecting the loss of performance in this earlier period.

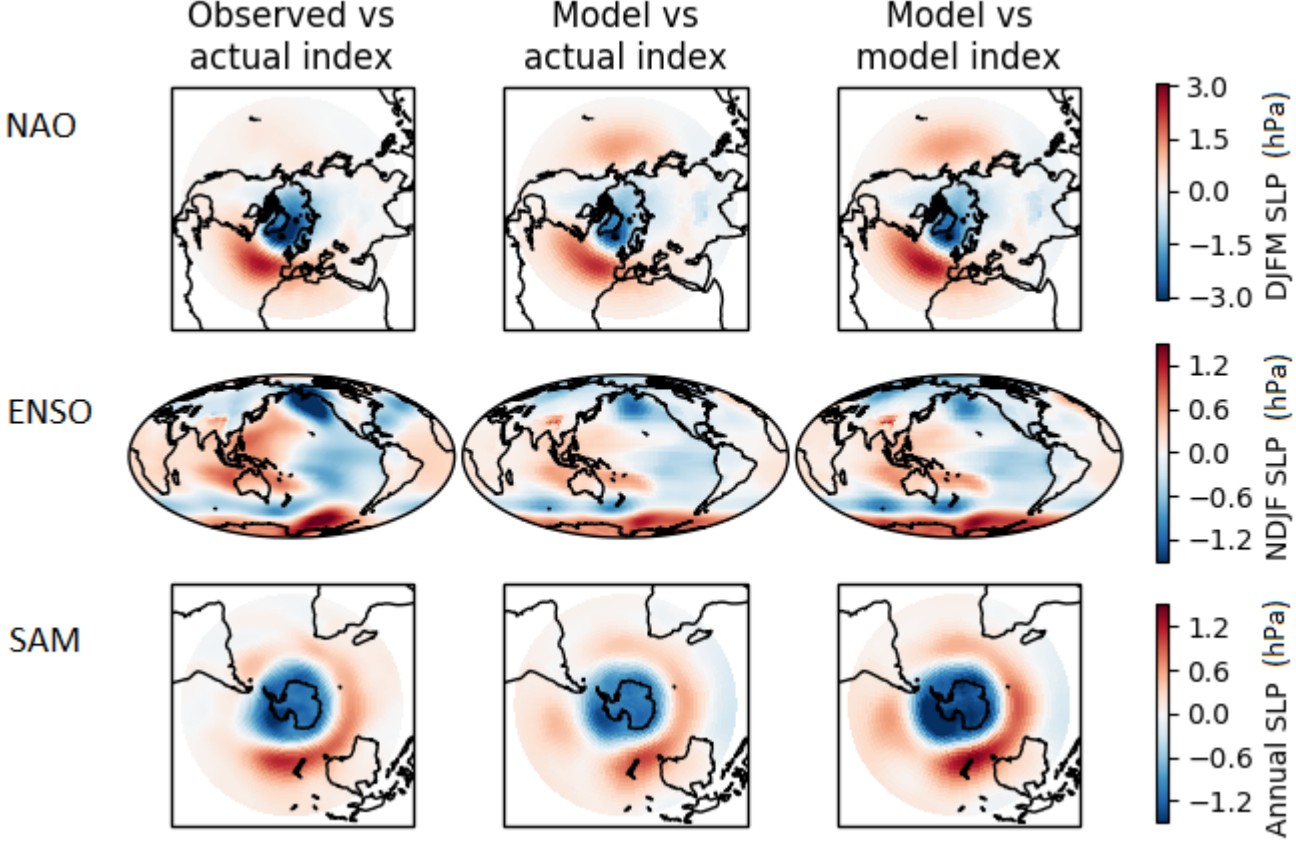

**Figure 4 - Regression between index and sea level pressure.** From top to bottom: NAO, ENSO, SAM. Left column, regression between the observed index and the observed spatial pattern of sea level pressure (from 20CRv3 Slivinski et al. 2021). Middle column, regression between the observed index and the DA model spatial pattern. Right column regression between the DA model index and its own spatial pattern. All data is first detrended. Regressions calculated over the period 1882-2008 except the SAM regressions which are calculated over the shorter period 1957-2008.


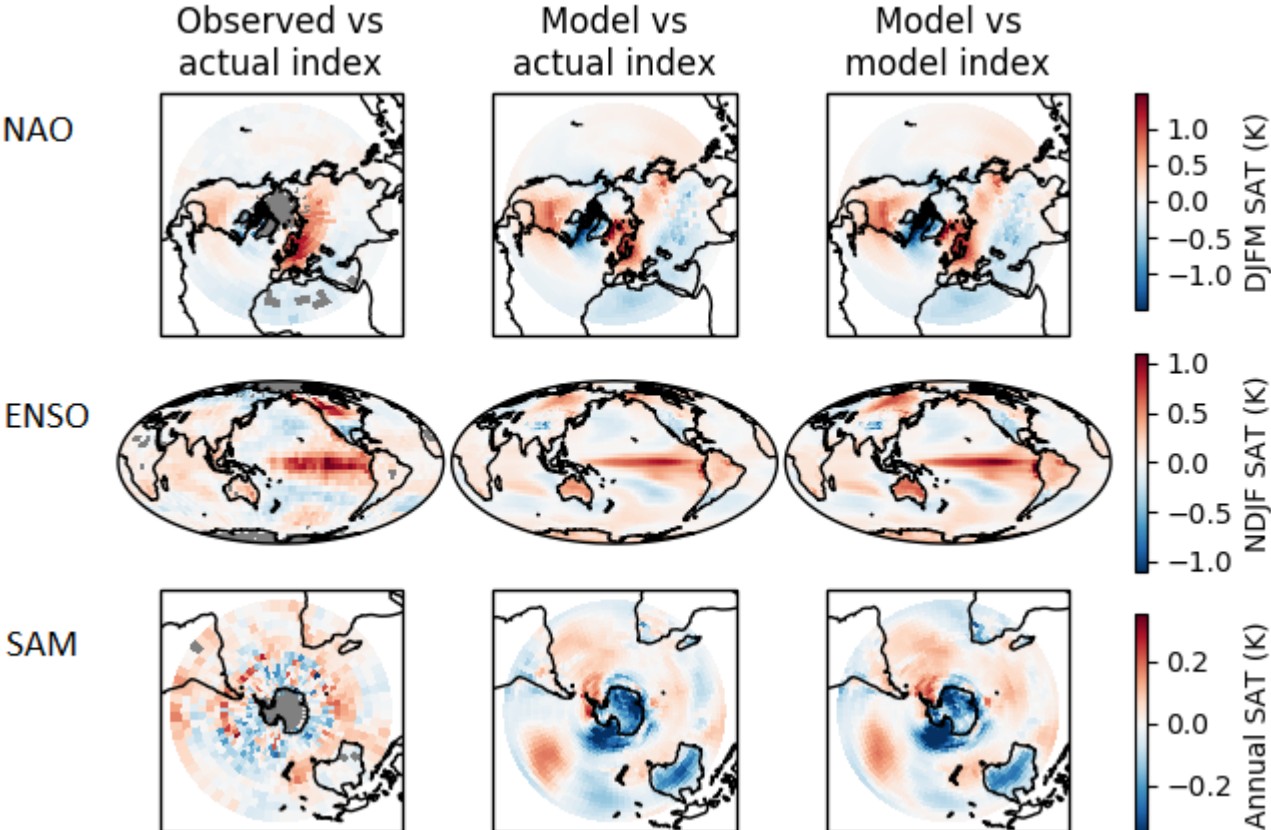

**Figure 5 - Regression between index and surface air temperature.** From top to bottom: NAO, ENSO, SAM. Left column, regression between the observed index and the observed spatial pattern of SAT (from HadCRUT5 Morice et al. 2021). Middle column, regression between the observed index and the DA model spatial pattern and. Right column regression between DA model index and its own spatial pattern. All data is first linearly detrended. Regressions calculated over the period 1882-2008 except the SAM regressions which are calculated over the shorter period 1957-2008 due to data availability. Grey indicates missing data.

## 3.2 Correlations of models with observed climate

In order to determine in which regions the particle filter is providing a statistically improved realisation of the observed climate compared to the forcing-only simulations we estimate skill by using correlation coefficients, as suggested in Goddard et al. (2013). Detrended observed anomalies are correlated with those simulated by the model with and without assimilation for each grid cell to determine where the assimilation is significantly improving the agreement. Because the particle filter results are a weighted mean over a number of particles (see supplementary Fig. S3) the internal variabilty not associated with the three assimilated modes will be reduced. Consequently comparing them directly to each of the forcing-only simulations could give



misleading results. To account for this, the average weights which are used in the particle filter experiment for the period analysed are calculated (see supplementary Fig. S4) and then applied at random to the 10 forcing-only simulations to create 100 different equivalently weighted-means. Results are considered significant if correlations calculated between the observations and the particle filter experiment are greater than 95% of the equivalent values calculated between the observations and the weighted means from the 10 forcing-only simulations. In order to focus exclusively on the skill gained

from including the assimilation, we calculate the difference between the correlation calculated for the assimilated experiment and the mean correlation calculated for the non-assimilated forcing-only simulations (see Fig. 6). The correlations between the assimilated and non-assimilated model simulations and the observations are shown in the supplement (Figs. S5, S6).

Spatial patterns of correlation differences are shown in Fig. 6a and b for annual and boreal-winter (DJF) surface air temperature (other seasons, Fig. S8). In both cases a substatial fraction of the total number of grid squares displays significanly higher

correlations than would be expected due to chance. Improved correlations are highest in the central and eastern tropical Pacific due to the effect of ENSO and regions with strong connections to this mode of variability will show similar behaviour in both models and observations, for example the Indian ocean and tropical Atlantic (Klein et al. 1999). Outside of the tropics teleconnections to the ENSO variability are much weaker. Other significant increases in correlation (above that expected by chance) are seen in the northern extratropics, in the north Atlantic, Europe and Siberia, particularly in winter. This is the region

most strongly affected by the NAO (see Hurrell et al. 2003 and Figs. 4 and 5) which is assimilated for this period (December – March). To further investigate correlations over this region we have analysed results using a spatial European reconstruction (Luterbacher et al. 2004) over the full period of the experiment; 1782-2000 (Figs. 6c and d), in winter and in the annual mean. Correlations are found to be particularly strong over the northern part of Europe in winter, consistent with the effect of the NAO. The effect of the SAM on temperature is not so clear, partly due to limitations in observational coverage. However,

there are high and significant correlations in parts of the Southern Ocean particularly south of Australia and off the southern tip of South America, regions strongly effected by the SAM (see e.g. Fogt and Marshall 2020) and which have good agreement between models and observations in Fig. 4. Note that Figs. 6a-d use the infilled analysis product of HadCRUT5, but the main findings are insensitive to the infilling process (see Fig. S7).

Differences in how well the assimilated and forced simulations correlate to observed precipitation are shown in Figs. 6g and

h. Due to sparcity in observations the anlaysis is restricted to land only regions with a start date of 1950. As for temperature, significant improvements are mainly found in the tropics particularly in the Pacific, but also in Europe and southern Australia consitent with the influence of the NAO and the SAM.

One striking aspect of Fig. 6 is that, even for temperature, the assimilation only significantly improves the correlations for about a third of the globe at interannual timescales, with much of the significantly correlated areas in the tropics. This is an

indication of the relative importance of the three modes of variability assimilated here compared to that of other components of internal variability. At higher latitudes the atmosphere's intrinsic variability is much higher, and although important, the relative role of the indices assimilated here can only explain a relatively small component of the total variability. This demonstrates both the limitations and strengths of this approach; while it will not give as accurate a realisation of past climate



as reanalyses which assimilate as much observed daily (or sub-daily) data as possible it can instead be used to assess the
relative role these three indices play in determining the climate.

One interesting aspect is how much variability these indices can explain on a decadal time-scale (filtering using a running 11-year mean). As Figs. 6c and d show, the most prominent area of improved correlations on these longer time-scales is the North Atlantic, in the sub ploar gyre region, which is an area that has previously been identified as one with a particualarly strong link to the NAO, and which could play a key role in driving ocean circulation (see e.g Zhang et al. 2019 for a review). There
are also significant correlation improvements in surface temperature on a decadal scale in the Pacific and Indian Ocean, over northern Eurasia, Africa and in the Southern Ocean. Due to the relatively short period analysed, results for precipitation on a decadal scale are very noisy (Figs. 6i and j). However they do show an increased agreement in the tropical Pacific and in Europe, consitent with the results for surface temperature.







**Figure 6 – Improvement in temperature and precipitation annual and NH winter (DJF) correlation between models and observations due to assimilation** *(The difference in correlation between observations and the model simulations with and without assimilation. a) and b) Temperature correlations using HadCRUT5* (Morice et al. 2021) *for period 1882-2008. c) and d) decadal (11-year running mean) temperature correlations using HadCRUT5 for period 1882-2008. e) and f) Correlation using a temperature reconstruction* (Luterbacher et al. 2004)*, for the period 1782-2000. g) and h) Correlation using GPCC precipitation dataset (*Schneider et al. 2016*) for the period 1950-2008.(i) and (f) Decadal (11-year running mean) precipitation correlation using GPCC for the period 1950-2008. Stippling indicates where correlation is bigger than the 95th percentile from the weighted means of the 10 forcing-only model simulations. The number at the top right of every plot indicates the percentage of grid cells which are stippled. If only due to uncorrelated random variability a value of 5% would be expected.*

## 3.3 Large-scale mean climate

Results for correlation against observed large scale average temperature support the spatial correlations discussed above. The agreement is significantly improved by the assimilation in the global mean and in the tropics and SH extra-tropics (Fig. 7). Temperature in the northern extra-tropics show an improvement in the particle filter during the second half of the century (particularly north of 35N) with larger correlations than in any simulation without data-assimilation, but not during the first half of the century, with values particularly low in the period 1900-1940 and no statistical overall improvement for the period as a whole. A similar result is found in correlations over Europe (Fig. 7h) with very low values at the start of the 20th century. As the NAO has a particularly strong influence on Europe (Hurrell et al. 2003) and the DA model follows the observed NAO closely throughout this period, these weak correlations suggest the effect of the NAO on Europe and northern high latitude regions is not constant through time, although uncertainties in observations could also contribute. Other studies (Weisheimer et al. 2019) have also found non-stationarities in the skill of initialised climate models to predict the NAO with higher values since the 1970s and lower in the mid-20th century. In general, relatively small improvements in correlation of temperature at large scale and over long time periods are expected because at these scales the effect of external forcing will dominate and much of the discrepancy between simulated and observed climate is likely to be due to errors in the simulated forced response. This is particularly true for higher latitudes where previous studies have found that despite a strong regional influence, the NAO only weakly affects large scale temperatures (Iles & Hegerl 2017).

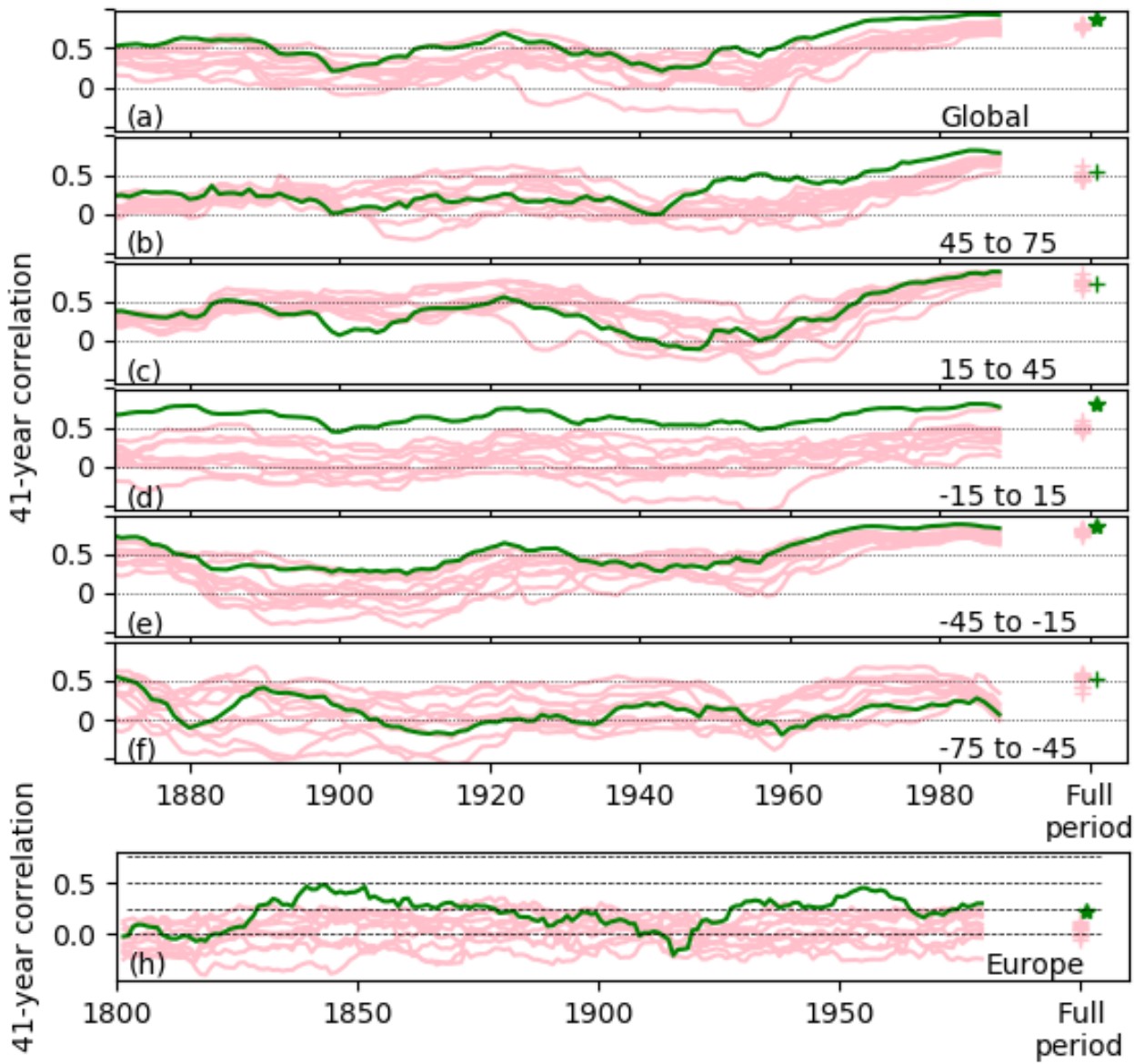

**Figure 7 Annual and DJFM 41-year sliding correlations for zonal-mean air temperature.** *Correlations are calculated for zonal mean temperatures for latitudes given in the bottom right of each panel between HadCRUT5 and* forcing-only *simulation (pink) and data-assimilation simulation (green). Correlations over the full period on right of plot, if correlation greater than all 10* forcing-only *simulations the green symbol is plotted as a star (instead of a plus).*

Previous studies, for example Friedman et al. (2020) and Hegerl et al. (2018) have found that observed decadal temperature variability in the southern hemisphere, particularly the extra-tropics, is far greater than that simulated by the latest climate models, and is outside the CMIP5 ensemble spread. As we are assimilating the key modes of variability for this region (the





ENSO and SAM), it is an interesting question to investigate whether our DA model simulation can offer insights into this
discrepancy. Time series of surface air temperatures averaged over zonal bands, including the SH extra-tropics are shown in
Fig. 8. As expected, the observed variability over 65 to 30S is outside the range of the forcing-only simulations. Assimilation
does little to improve this situation, suggesting that it is neither the SAM nor the ENSO variability which is responsible for the
discrepancy. However, this could partially be due to possible deficiencies in the model's pattern of the SAM (see Figs. 4 and
5). Other intriguing possibilities are limitations in model variability in the Southern Ocean (Hyder et al. 2018, Beadling et al.
2020), as well as possible biases in SST measurements such as those found in the NH Atlantic and Pacific (Chan et al. 2019),
and using coastal weather stations (Cowtan, et al. 2018).

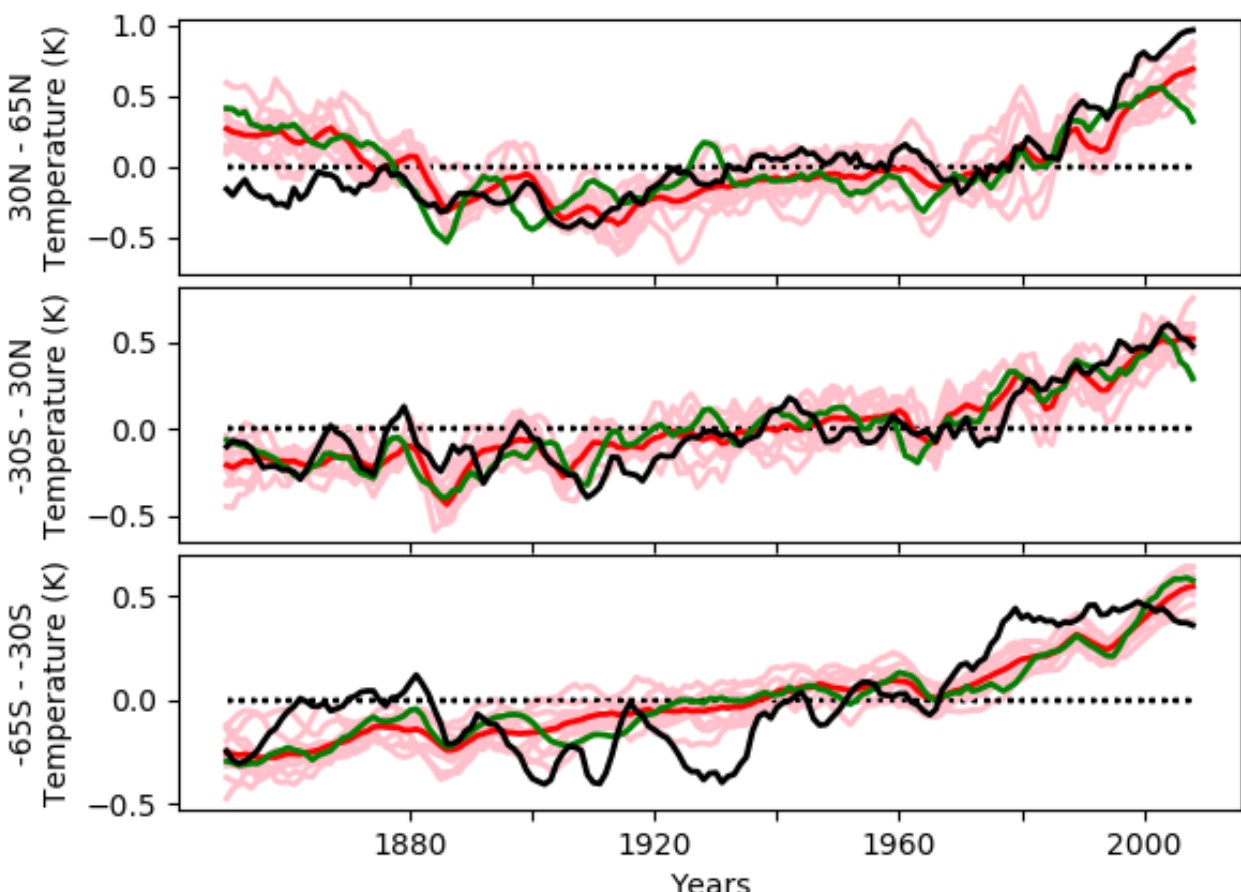

**Figure 8 – Mean annual surface air temperature averaged over three latitudinal bands** *Assimilated model (green), 10*
*forcing-only simulations (light pink), forcing-only ensemble mean (red), observations (HadCRUT5; black). All timeseries*
*filtered using a 5-year running mean.*



### 3.3 Temperature response to large volcanic eruptions

Volcanic eruptions have an important role on annual as well as decadal climate (Robock 2000). However, the global mean
temperature response has been found to be stronger in climate models than in observations (Chylek et al. 2020, Bindhoff et al
2013). It has been suggested that this discrepancy could be explained, at least partially, by ENSO events which coincide with
all major volcanic eruptions during the instrumental era (Lehner et al. 2016), although the role of the volcanic eruption in
triggering the occurrence of an El-Nino is still disputed. Studies of the last millennium which rely on proxy evidence offer
mixed results from a longer term perspective with some recent studies, typically based on tree-ring evidence supporting an El-
Nino like response in the year following an eruption (e.g. McGregor et al. 2010, Li et al. 2013), while studies based on coral
data from the ENSO region itself do not find any significant relationship (Tierney et al. 2015, Dee et al. 2020). Models of
different complexities also simulate an El-Nino response, although they disagree on the underlying mechanisms (Ohba et al.
2013, Khodri et al. 2017, Predybaylo et al. 2020, Maher et al. 2015, Hermanson et al. 2020). Given that the data-assimilation
simulation has been designed to follow the observed ENSO index it provides an ideal test bed for investigating the effect of
the ENSO on the climate after volcanic eruptions. Although it cannot provide any evidence to address the question of whether
a forced link exists between the eruption and the ENSO state.

To investigate the temperature response following volcanic eruptions we employ a commonly used technique called an epoch
analysis (e.g. Hegerl et al. 2003). This involves averaging the response across multiple eruptions in order to reduce internal
variability to focus on the forced response. Here we choose the 5 strongest tropical eruptions which coincide with the period
of instrumental observations: Krakatoa (August 1883), Santa Maria (October 1902), Agung (March 1963), El Chichon (April
1982), Pinatubo (June 1991). The global mean surface air temperature and Nino3.4 index are calculated for the simulated and
observed climate for 60 months after each eruption occurred relative to a pre-eruption mean (calculated as the mean over the
60-months directly before the eruption) and the results for all 5 are averaged together. Fig. 9 shows the results for the epoch
analysis for the last 5 major tropical eruptions. The ENSO index concurrent with the eruptions is on average slightly positive
when the volcanic eruptions occur and becomes increasing positive for the year after the eruptions. This behaviour is captured
remarkably well in the DA simulation but not by any of the forcing-only simulations. Based on the results from the forcing-
only simulations one would conclude that the model overestimates the response to volcanic eruptions, since all model
simulations have larger cooling following the eruption than that observed. Assimilating ENSO, however resolves this clear
mismatch, showing that, at least for this model, if a simulation has the correct ENSO evolution the temperature response is
very close to that observed.





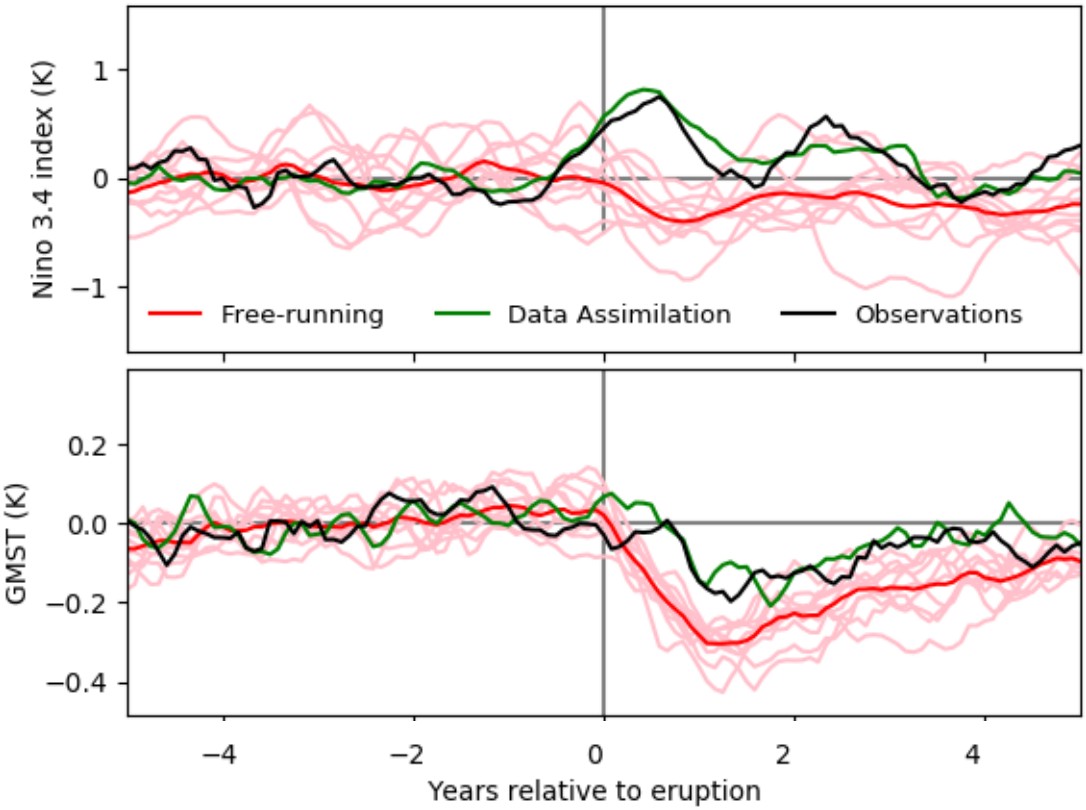

**Figure 9 – Response to 5 tropical volcanic eruptions.** *Epoch analysis for Top -observed Nino3.4 SST anomaly index. Bottom*
*- global mean surface air temperature. Observations are HadSST3, and HadCRUT4.6, all data masked to the observations.*
*Both variables are plotted as anomalies from the mean of the 5-years before the eruption date.*

## 4 Discussion and Conclusions

In this study, we have introduced a new experimental set-up, based on existing data-assimilation techniques, to produce a
near-continuous and near free-running model simulation with a realisation of the ENSO, NAO and SAM similar to that which
actually occurred. We have demonstrated that this method adequately captures the assimilated modes and offers an improved
realisation of the climate above that of free-running simulations without assimilation in several key regions in both temperature
and rainfall. This includes: annual and winter mean surface temperature over much of the northern extratropics, large parts of
tropical surface temperature, and tropcial and European rainfall. On a decadal scale improvements in skill are found over much
of the tropical Pacific, over some parts of the Southern Ocean and in the sub-polar gyre region of the northern Atlantic. Decadal
forcast experiments have shown that the NAO (Smith et al. 2020) and ENSO (Barnston et al. 2019; Dunstone et al. 2020) are
predictable on seasonal to annual timescales. This analysis highlights where a prediction system has the potential to better
reproduce observed trends, if the prediction systems were capable of following the correct realisation of these modes.



Interestingly our results also display some evidence of non-stationarity, with the assimilation improving agreement to observations more in some periods than others, suggesting that the skill in decadal forecasts could vary through time.

Past modelling studies have shown that the observations in the southern extratropics are far more variable on the decadal scale than model simulations. By assimilating the ENSO and the SAM we can show that getting the evolution of these modes of variability closer to reality does not help to explain this discrepency. This suggests that the difference must have an alternative origin, such as observational errors or other model deficiencies. Outside this region though, there is no major discrepancy in Surface air temperature, providing support for the ability of climate model's to reproduce observed climate evolution.

Consistent with most models, the HadCM3 simulations analysed here show too much cooling following volcanic eruptions over the last 150 years. Correctly assimilating ENSO results in simulated cooling that is consistent with the observations. This demonstrates that the model response is not necessarily too strong, rather that the forcing-only simulations do not capture the correct ENSO response. This conclusion highlights that relying on model simulations which do not capture the observed ENSO variability could lead to misleading conclusions.

We therefore consider this use of a particle filter to be a promising experimental design which could be developed further in the future to investigate the causes of past variability or the effect of potential future changes in key circulation modes using a storyline approach. We note that while we have chosen to assimilate three modes of variability in this study, it would be possible to assimilate any number of them. Assimilating fewer modes would reduce the degrees of freedom for the filter and hence allow an improved performance for that mode or alternatively the ability to use less particles, while assimilating more

has the potential to reduce the performance of the filter unless the number of particles are also increased.

**Acknowledgements**

The model simulations and AS, GH, MM and ST were funded by the ERC funded project TITAN (EC-320691) and made use of the resources provided by the Edinburgh Compute and Data Facility (ECDF) (http://www.ecdf.ed.ac.uk/). AS and GH were further funded by NERC under the Belmont forum, Grant PacMedy (NE/P006752/1), the NERC projects GloSAT

(NE/S015698/1) and Vol-Clim (NE/S000887/1) and AS received funding from a Chancellors fellowship at the University of Edinburgh. DMS was supported by the Met Office Hadley Centre Climate Programme funded by BEIS and Defra. HG is research director at the Fonds de la Recherche Scientifique (F.R.S.-FNRS). MHE was supported by the Australian Research Council (SR200100008).



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
