# Peer review of "Quantifying the contribution of forcing and three prominent modes of variability on historical climate"

_Climate of the Past, 2022_

## Author Comment (AC1)

*Reviewer #1*

*Thank you for your constructive and positive review. We will discuss each of your specific points below, with our replies written in bold.*

Major comments:

Line 104 – how does the choice to stop the simulations on April 1 affect the results? Would the results differ if you used a different month?

*This is an interesting question. We decided to stop the simulation and to perform the assimilation step on April 1 because the majority of the indices assimilated are winter indices (in particular the NAO and to a lesser extent the ENSO), so it seemed most logical that this was done after the end of winter period assimilated. Thus the simulations are restarted from initial conditions which best agreed with the winter period which had just occurred. In the revised manuscript we will ensure that this point is clear in the text. There are a number of different ways the experiment could have been set up but due to computational limitations we were not able to explore this fully.*

Figures – in generel the use of red and green is not good for colorblind people. Please update the colorschemes. There are online tools to check whether a Figure is readable if someone is colorblind.

*Good point – thanks for bringing this to our notice – all figures this applies to will be updated, and the green line representing the assimilated simulations will be replaced by a blue line in all instances (including the supplementary information).*

Figure 3 – the red and green and black lines are hard to distinguish – perhaps the use of dots or dashes for those plotted on top of others would help.

*We thank the reviewer for this suggestion – but after careful consideration we do not think that dotted or dashed lines will clearly show the variability. We will however change the green line to blue – in response to the comment above – and will try to ensure that this figure is as clear as possible.*

Figure 4/5 – I recommend adding panels where differences are taken – this would be much easier to interpret

*The purpose of this figure was not only to show how similar the models and the observations are but also to show what the pattern is. The pattern itself is crucial to understand the performance of the experiments as they show which regions you could expect to be influenced by the assimilation. We therefore propose keeping the figure as is, but we will also add a figure showing the difference in the patterns to the supplement.*

Figure 6 – Why are you looking at boreal winter alone?

*For the main figures we concentrate on boreal winter as this is the season which contains the most assimilated data, we will add a short statement to the text to make this clear. The other seasons are shown in figure S8.*

The bottom two panels are hard to interpret – would smoothing help?

*This is a good idea – but given that the figure is already effectively smoothed by the use of a running correlation – we consider that this would make the figure harder to interpret so we would like to keep the figure as it is.*

Figure 7 -

The caption says this is Annual and DJFM but I only see one result.

*Thank you for pointing this out – we will update the caption.*

There is currently no panel (g)

*Thank you for spotting this– we will update the labelling.*

**Minor comments:**

Line 35 – should also cite the original paper by Hawkins and Sutton, 2009: https://journals.ametsoc.org/view/journals/bams/90/8/2009bams2607_1.xml

*This will be included*

Line 45 needs citation:

Some options:

https://www.nature.com/articles/s41558-020-0731-2

https://link.springer.com/article/10.1007/s00382-010-0977-x

*These will be included - thanks*

Overview of large ensemble literature – could be useful for lines 40-45:

https://esd.copernicus.org/articles/12/401/2021/

*We will add this citation to what was line 40.*

Line 80 – first thought is can we really trust data from 1781 – I see later you use reconstructions, this is great but perhaps needs to be mentioned earlier on line 80.

*We will add the following lines to the introduction – so hopefully this point will be clear from early on "For the start of the simulations the modes assimilated will mainly rely on proxy reconstructions with instrumental observations used later when it becomes available"*

Line 118 – remove repeated word "schematically"

*This will be done, thanks*

Line 225 – tell the read which color this is in brackets for ease of interpretation

*This will be done, thanks*

Line 299 – could this lack of variability in the Southern Ocean be due to the coarse resolution of the model?

**Yes this could be one possibility and note that this was suggested by Beadling et al (2020). We will highlight this by adding the wording "potentially due to the relatively coarse resolution"**

Section 3.3 either be clear that you refer to only tropical eruptions or add citations for high-latitude eruptions: some are as follows

https://www.pnas.org/doi/10.1073/pnas.1509153112

https://esd.copernicus.org/articles/12/975/2021/

https://www.cesm.ucar.edu/projects/community-projects/LME/publications/Stevenson-JClimate-2016.pdf

*We agree that this would be good to clarify so we will add a sentence on high latitude eruptions with some of the references you suggest cited.*

Line 316 – another possible citation

https://www.nature.com/articles/s41467-022-28210-1

**Thanks we will add this as well**

Section 3.3 – there is a review on this topic:
https://agupubs.onlinelibrary.wiley.com/doi/chapter-epub/10.1002/9781119548164.ch12

**Thanks this is a useful reference and will be added**

Line 354: Does this relate to these results:
https://agupubs.onlinelibrary.wiley.com/doi/full/10.1002/2015GL066608

Line 363 – can you say why?
**The question of why this model (like the majority of models) does not capture the correct ENSO response is an important question but we feel it is outside the scope of this paper.**

---

## Author Response (AR1)

We thanks both reviewers for the helpful reviews which we feel has strengthened the paper. We will discuss each of the specific points below, with our replies written in bold.

*---------------------------------------------------------------------*

*Reviewer #1*

*Thank you for your constructive and positive review.*

Major comments:

Line 104 – how does the choice to stop the simulations on April 1 affect the results? Would the results differ if you used a different month?

*This is an interesting question. We decided to stop the simulation and to perform the assimilation step on April 1 because the majority of the indices assimilated are winter indices (in particular the NAO and to a lesser extent the ENSO), so it seemed most logical that this was done after the end of winter period assimilated. Thus the simulations are restarted from initial conditions which best agreed with the winter period which had just occurred. In the revised manuscript we have added a sentence to ensure that this point is clear.*

*"The assimilation step was chosen to occur on April 1st, so the simulations are restarted from initial conditions which best agree with the boreal winter which has just occurred (the period in which the most information is assimilated)."*

*There are a number of different ways the experiment could have been set up but due to computational limitations we were not able to explore this fully.*

Figures – in general the use of red and green is not good for colorblind people. Please update the colorschemes. There are online tools to check whether a Figure is readable if someone is colorblind.

*Good point – thanks for bringing this to our notice – all figures this applies to have been updated, in particular the green line representing the assimilated simulations will be replaced by a blue line in all instances (including the supplementary information).*

Figure 3 – the red and green and black lines are hard to distinguish – perhaps the use of dots or dashes for those plotted on top of others would help.

*We thank the reviewer for this suggestion – but after careful consideration we do not think that dotted or dashed lines will clearly show the variability. We have however changed the green line to blue – in response to the comment above – and will try to ensure that this figure is as clear as possible.*

Figure 4/5 – I recommend adding panels where differences are taken – this would be much easier to interpret

*The purpose of this figure was not only to show how similar the models and the observations are but also to show what the pattern is. The pattern itself is crucial to understand the performance of the experiments as they show which regions you could expect to be influenced by the assimilation. We therefore propose keeping the*

*figure as is, we have added a figure showing the difference in the patterns to the supplement (Fig. S5).*

Figure 6 – Why are you looking at boreal winter alone?

*For the main figures we concentrate on boreal winter as this is the season which contains the most assimilated data, we have added a short statement to the text to make this clear.*

*"the focus is on boreal winter as this is when most data is assimilated, results for other seasons are shown in the supplement , Fig. S9"*

The bottom two panels are hard to interpret – would smoothing help?

*This is a good idea – but given that the figure is already effectively smoothed by the use of a running correlation – we consider that this would make the figure harder to interpret so we would like to keep the figure as it is.*

Figure 7 -

The caption says this is Annual and DJFM but I only see one result.

*Thank you for pointing this out – we have updated the caption.*

There is currently no panel (g)

*Thank you for spotting this– we have updated the labelling.*

**Minor comments:**

Line 35 – should also cite the original paper by Hawkins and Sutton, 2009: https://journals.ametsoc.org/view/journals/bams/90/8/2009bams2607_1.xml

*This has been included*

Line 45 needs citation:

Some options:

https://www.nature.com/articles/s41558-020-0731-2

https://link.springer.com/article/10.1007/s00382-010-0977-x

*These have been included - thanks*

Overview of large ensemble literature – could be useful for lines 40-45:

https://esd.copernicus.org/articles/12/401/2021/

*We have added this citation to what was line 40.*

Line 80 – first thought is can we really trust data from 1781 – I see later you use reconstructions, this is great but perhaps needs to be mentioned earlier on line 80.

*We have added the following lines to the introduction – so hopefully this point will be clear from early on "For the start of the simulations the modes assimilated will mainly rely on proxy reconstructions with instrumental observations used later when it becomes available"*

Line 118 – remove repeated word "schematically"

*This has been done, thanks*

Line 225 – tell the read which color this is in brackets for ease of interpretation

*We are unfortunately unsure what the reviewer is referring to here, as we cannot see how this applies to the specified line number.*

Line 299 – could this lack of variability in the Southern Ocean be due to the coarse resolution of the model?

**Yes this could be one possibility and note that this was suggested by Beadling et al (2020). We have highlighted this by adding the wording "potentially due to the relatively coarse resolution"**

Section 3.3 either be clear that you refer to only tropical eruptions or add citations for high-latitude eruptions: some are as follows

https://www.pnas.org/doi/10.1073/pnas.1509153112

https://esd.copernicus.org/articles/12/975/2021/

https://www.cesm.ucar.edu/projects/community-projects/LME/publications/Stevenson-JClimate-2016.pdf

*We agree that this would be good to clarify so we have added a sentence on high latitude eruptions with some of the references you suggest cited.*

Line 316 – another possible citation

https://www.nature.com/articles/s41467-022-28210-1

**Thanks we have added this**

Section 3.3 – there is a review on this topic:
https://agupubs.onlinelibrary.wiley.com/doi/chapter-epub/10.1002/9781119548164.ch12

**Thanks this is a useful reference and has been added**

Line 354: Does this relate to these results:
https://agupubs.onlinelibrary.wiley.com/doi/full/10.1002/2015GL066608

Line 363 – can you say why?

**The question of why this model (like the majority of models) does not capture the correct ENSO response is an important question but we feel it is outside the scope of this paper.**
* * *
**Reviewer #2**

**We are grateful to the reviewer for their positive comments. The reviewer wrote:**

The only concern I have is that the verification of the DA only includes the correlation metric, which by its nature does not provide any information about bias. I think the authors ought to provide, at least in the supplement, some verification metric that incorporates bias, such as 'bias' or 'mean absolute error' or 'continuous-ranked probability score'. So I am suggesting a verification just like Fig 6 but for an additional metric. This will allow for a more complete assessment of the limitations of the DA product.

**This is a good idea and we agree that this suggestion represents a useful test of the model and one which will improve the interpretation of our results. We have therefore added a figure (Fig. S10) showing mean absolute error, and some text to the main paper. The conclusions support the main correlation results, with improvements in most of the same regions, although the bias metric highlights areas of higher variability.**

---

## Author Response (AR2)

**We thank the editor for their very helpful comments which will lead to improvements in the presentation of our results. We detail how we have responded to each of these comments below (in bold).**
* * *
Figure 2:
* The small boxes with "NAO", "ENSO" and "SAM" are distracting in the way that the appear on the time line. Suggest instead moving these labels to the left side of each of the time line bars.

* for proxy SAM the December "D" indicator is missing off of the months given.

* I find the way that the black lines marking the transition from proxy to observed data continue into the white space to be distracting. Could these please be restricted just to within the time line box that they refer to.

**We have made the changes as suggested and agree that this has improved the figure. Thank you.**

Figure 3:
"Reconstructions (orange)" should be "Reconstructions (red)"

**We have changed the caption accordingly (although note the reconstructions are actually plotted in brown).**

* To what extent might the larger magnitude of reconstructed SAM changes compared to observations (as evident in the 3rd row of figure 3) be related to methodology in the way the observed annual average SAM index is calculated? The Marshall SAM index page gives an annual (Jan-Dec) SAM index that is calculated by first calculating annual mean zonal MSLP and then constructing the SAM. Because you are using a different set of months to calculate the annual SAM I suspect that your observed annual SAM is instead calculated by averaging across the monthly SAM index. This results in a large difference in the magnitude of the SAM index. The SAM index calculated by Marshall by first taking zonal mean annuals has a much larger (though theoretically dimensionless because of the normalisation step in calculating the SAM index) magnitude of variability than an annual SAM value calculated across the monthly SAM index. The Abram et al 2014 SAM index used that former version of annual SAM index to scale the proxy data, hence resulting in a large magnitude of variability in the proxy, which I don't think is comparable to the annual SAM index you use for the observations. See Nicky Wrights paper for details: https://cp.copernicus.org/articles/18/1509/2022/cp-18-1509-2022.html

**Thank you for this very thoughtful suggestion. We have double checked the analysis and are confident that it is OK as it is. To be consistent we are actually plotting the annual SAM index (Jan-Dec) for everything (model simulations, observations and reconstructions). We appreciate this is unclear in the figure since this is not what we are assimilating during the final period and have updated the caption to make this clear.**

**We also have realised that the description of what SAM index was assimilated was also not clear. The metric initially assimilated was the annual mean SAM as**

defined in Abram et al. After 1957 the SAM assimilated was the annual mean of the monthly values of the Marshall index (April - March). We have added to the main text so it now reads.

*Given that monthly data exists, the mean of the period April through to March is used to coincide with the assimilation time-step, and the metric for this period is calculated as the mean of the monthly values.*

**We hope this is all clear now.**

(let me know if you would like the rescaled version of the Abram 2014 SAM Index using annual average instrumental target calculated from monthly SAM data, as presented in Nicky's paper)

**We do not think this is needed but we are grateful for your offer of help.**

Figure 5:
On the SAM figure for "observed v actual index" (lower left panel) it seems odd to have Antarctica shaded grey with no data but SAT data showing over the whole of the Southern Ocean. Could ERA5 be used instead to provide coverage over Antarctica? Or could you combine HadCRUT5 with the spatial temperature dataset for Antarctica from Steig et al., 2009, Nature (doi:10.1038/nature07669)?

Note, I then also am confused as to how Figure 6 shows data using HadCRUT5 temperature correlations over Antarctica, whereas the Antarctic continent was shaded grey for missing data in Figure 5?

**Thank you again, for an excellent observation. The difference is due to an inconsistency in our analysis. The figures used different versions of the HadCRUT5 dataset – Fig 5 used the infilled dataset which has more coverage and Fig 6 the un-infilled version of the dataset. We have updated figure 6 to use the infilled dataset and have updated all relevant figure captions to make it clear which version was used in each.**

Figure 6:
I think that it would be helpful to add % signs after each of the numbers given at the top right of each panel.

**Agreed – we have updated this figure to include this suggestion, as well as all relevant figures in the supplement.**

Figure 7 b-f:
It would be clearer if the zonal mean labels gave degree symbols and N or S (e.g. "75°S to 45°S" rather than "-75 to -45")
Figure 8:
y-axis labels: remove negative signs as you indicate hemisphere already with N and S.

**Agreed – we have made these changes. Thank you.**